# Factors Associated with the Psychological Health of Caregiving Older Parents and Support from Their Grown Children: Results from the China Health and Retirement Longitudinal Study

**DOI:** 10.3390/ijerph17020556

**Published:** 2020-01-15

**Authors:** Liping Fu, Yuhui Wang, Lanping He

**Affiliations:** 1College of Management and Economics, Tianjin University, Tianjin 300072, China; lpf3688@126.com (L.F.); wangyuhui2018@tju.edu.cn (Y.W.); 2Center for Social Science Survey and Data, Tianjin University, Tianjin 300072, China

**Keywords:** caregiving older parents, CHARLS data, factors, psychological health, stepwise decreasing logistic regression

## Abstract

In China, older parents have become an important source of childcare for their grown children since 2010. However, caring for grandchildren may affect older parents’ psychological health (PH) in both positive and negative ways. Using the method of stepwise decreasing logistic regression, this study aimed to assess the factors associated with PH and support from grown children among caregiving older parents (400 respondents) based on the public panel data of the China Health and Retirement Longitudinal Study (CHARLS). The findings showed that being male (X1, OR = 1.661 (95% CI 1.066–2.590)), being literate (X4, OR = 2.129 (95% CI 1.369–3.309)), and expecting long-term care in the future from their grown children (X6, OR = 2.695 (95% CI 1.736–4.185)) were significant factors associated with PH. Therefore, in such an aging society, we should not regard older parents as a “burden”; we should recognize and appreciate their contribution to caregiving. As family and children, it is important to give older parents the necessary economic and emotional support to maintain their psychological health in the meantime.

## 1. Introduction

Since 2010, the one-child policy of China has gradually opened up, the number of second births has increased [1,2], and an increasing number of elderly people have had to take care of their grandchildren due to the inadequacies of kindergarten classes and the insufficiency of China’s current child-care system [3,4,5]. The results of the China Health and Retirement Longitudinal Study (CHARLS) showed that nearly 60% of the elderly needed to take care of their grandchildren in 2015 [6]. According to the data of Shanghai Academy of Sciences, 84.6% older parents tried to help their grown children by caregiving for their babies in Shanghai. However, under great economic and survival pressure, many grown children are unable to give their older parents enough support and understanding in the process of caring for the babies of the family. The heavy caregiving tasks have affected older parents’ psychological health (PH) in China [7,8,9,10].

In previous studies, it is usually noted that there are some health benefits of caregiving from older parents and those benefits are recognized by grown children [11]. It can make up for negative feelings of older empty nesters, lead to a happier life, and improve their quality of life [12,13,14,15,16]. This activity is significant in many countries [17]. Older parents played a more important role in the family and helped a lot for grown children [18]. If older populations did their grown children a favor by caregiving, grown children would have more time, pay more attention to their career, and get better development. Thus, in return, grown children’s care for the elderly and time with family would increase; this is how family works [19,20]. Then, these supports from family are the most beneficial things to the older parents, especially for older populations who choose home-based care. The benefits of receiving psychological support from grown children are fully mediated by parents’ satisfaction with their children [21,22,23,24,25,26]. Family has become the most important place for older parents.

In addition, some past studies showed that caring for grandchildren increased the psychological pressure of older parents, took up a great deal of time, and caused them to neglect their own health [27,28]. In the aging state of caregiving older parents, not only do they need to take care of their grandchildren, but also themselves [29,30]. If older parents live with their grown children, they also need to consider their children’s living habits, working conditions, and physical health. However, the living habits and self-care of the elderly are not consistent with the life needs of their grown children and grandchildren, and some of them are even completely opposed [31,32,33,34]. Especially in China, sometimes older parents need to sacrifice themselves in order not to cause trouble to their grown children, which increases the risks to their PH [35,36]. Moreover, the physical condition and pension of older parents in China are worse compared with developed countries, and they are more dependent on their grown children [37,38,39,40]. However, in the face of severe life challenges, such as rising house prices and employment difficulties, many adult children in China are unable to provide their parents with sufficient economic and emotional support.

According to the above literature, taking care of grandchildren has positive and negative effects on the PH of older parents. For most caregiving older parents, both effects are likely to occur. Older parents can be regarded as either a “support” or a “burden”, and how they are seen mainly depends on which kind of effect is stronger. The factors associated with the PH among Chinese caregiving older parents need to be studied further.

## 2. Materials and Methods

### 2.1. Data and Method

The analysis was based on public panel data collected as part of the China Health and Retirement Longitudinal Study (CHARLS), which was started officially by the Ministry of Education in 2011 and initially operated by Beijing University (BJU). It was designed to provide comprehensive information about the health and factors associated with the health of the older Chinese populations. Our research center was a cooperative unit with BJU. The questionnaire, ethical proof, investigation, and database are available on the website [41]. After application, any researcher can download it. For the purpose of this study, we selected 400 respondents who provided care for babies of their grown children.

Stepwise decreasing logistic regression was performed to explore factors associated with PH, using statistical software STATA 15.0. Final significant factors were obtained by gradually decreasing the non-significant variables, starting with the least significant variables (which had the highest *p*-value). In this way, the errors caused by multicollinearity, where there are very high intercorrelations or inter-associations among independent variables, could be avoided, and the change in the confidence intervals of statistically significant variables could also be observed. For the results of regression, a variable would be a protective factor when the value of the odds ratio (OR) was greater than 1 and a risk factor when the value of the OR was below 1.

### 2.2. Description of Variables

As reported in Table 1, the dependent variable is “Good psychological health”. Its criterion consisted of a corresponding scale from CHARLS 2015, consisting of 10 specific evaluation indicators (such as whether there is depression or not). Each indicator was reviewed with a self-anchoring scale with a range from 0 to 5. Older parents who were considered to be in good psychological health (value = 0) needed to reach the first three grades (the possible answers from respondents were excellent, very good, and good). The validity of the scale index was considered acceptable based on the result of Cronbach’s α coefficient (α = 0.827).

In terms of the independent variables, also shown in Table 1, “Gender ” (X1, Male = 0, Female = 1), “Age” (X2, Age under 65 years old = 0, Age over 65 years old = 1), “Residence” (Live in the urban area = 0, Live in the village = 1), “Educational status” (X4, Literate (primary school or above) = 0, Illiterate = 1), “Marital status” (X5, Married = 0, Another situation = 1), “Expectations of long-term care from their grown children in the future” (X6, Yes = 0, No = 1), “Educational status of grown children” (X4, Literate (primary school or above) = 0, Illiterate = 1), “Living place of grown children” (X8, Live with their older parents = 0, Not live with their older parents = 1), “Marital status of grown children” (X9, Married = 0, Another situation = 1), “Physical health of grown children” (X10, Good or better = 0, Poor or worse = 1), and “Economic status of grown children” (X11, At least own one house = 0, No house yet = 1) were collected as possible factors from CHARLS based on the previous study. The sample distribution is also summarized in the table.

## 3. Stepwise Decreasing Logistic Regression Results

As reported in Table 2, for the psychological health among caregiving older parents in China, being male (X1, OR = 1.828 (95% CI 1.137–2.940)), being literate (X4, OR = 1.810 (95% CI 1.122–2.921)), having expectations of long-term care from their grown children in the future (X6, OR = 2.707 (95% CI 1.735–4.222)), and having grown children with good physical health (X10, OR = 1.496 (95% CI 0.965–2.320)) were certified as significant protective factors based on the OR value (OR > 1).

We did not find any significant difference with respect to the age (X2), residence (X3), marital status (X5), educational status (X7), living place (X8), or marital (X9) or economic status (X11) of respondents’ grown children this time. Among all factors, the *p*-value of the older parents’ marital status (X5, 0.915) was the highest.

After dropping factor X5, logistic regression was performed again as the second analysis step. The results are summarized in Table 3. There were still four significant factors. The *p*-value of X7 (0.845) was then the highest.

X7 was then dropped, and logistic regression was employed again as the third analysis step. The results are summarized in Table 4. A *p*-value of X2 (0.729) was then the highest.

Then, in turn, X2, X8, X11, X9, and X10 were dropped, and logistic regressions were performed again as the last analysis step (results from the fourth step to the eighth step were showed in Table A2, Table A3, Table A4, Table A5 and Table A6 in Appendix A and all the step details were summarized in Table A1). The results are summarized in Table 5. “Being male” (X1, OR = 1.661 (95% CI 1.066–2.590)), “being literate” (X4, OR = 2.129 (95% CI 1.369–3.309)), and “expecting to receive long-term care in the future from their adult children” (X6, OR = 2.695 (95% CI 1.736–4.185)) were finally certified as three significant factors. In addition, we were able to determine the influencing ranking of the factors by individual degrees. The later the variables were deleted, the more important they were.

## 4. Discussion

In the face of the reality that an increasing number of older people are providing care for their grandchildren, the current study aimed to assess the factors associated with the psychological health of caregiving older parents.

The results show that being male and literate were protective factors. Men may have better physical strength for a great deal of caregiving activities. It could also be that perhaps men do not do as much caregiving as women and hence they reported better PH. Educated elderly tended to learn more to adapt to the process of caring for grandchildren [42]. Older parents tried to keep the connection between their grown children and them [43,44]. Moreover, education had other aging effects, including better employment, better pensions, and better living conditions. Children provided a greater sense of security to their older parents, such as having good physical health and a commitment to long-term care in the future. In such an aging society of China, grown-up children played an important role in parents’ aging state [2,8,10,12,29].

Compared with previous studies, this paper makes the following contributions. First, in addition to age, gender, and other common influencing factors of the elderly, this study also took into account the relevant factors of the adult children of older parents. It would help to identify the different impacts on the care of grandchildren by older people. Second, a stepwise decreasing logistic regression method was used to avoid the statistical error caused by the mutual influence of various factors. Third, this study empirically analyzed the inter-generational support among older parents and grown-up children, which provided evidence for the study of PH of the elderly.

There are some limitations in this paper. First, this study did not set up a control group to explore the factors associated with PH among the elderly who did not provide inter-generational care for grandchildren. Second, this study could not determine the specific economic and emotional support from grown-up children due to database limitations. Since this study was designed as a cross-sectional study, it was not possible to determine the time change of caregiving. Third, as 61.7% of participants were older than 65 years of age and 77% were married, they may provide caregiving for elderly spouse who have suffered from stroke or dementia. Previous studies reported that 34% of caregivers for dementia patients suffer from depression and 40% of caregivers for stroke suffer from depression [45,46]. This study did not report the prevalence of participants who need to provide caregiving to elderly spouses, which could be a confounding factor. Fourth, we dealt with opinions rather than validated measures to assess psychological distress. The longer the elderly took care of their grandchildren, the greater the potential threat was to their health [3,5,9,10,11,16,28,31]. All these aspects are worth further study in the future. Through this balance between “ask for support to take care of grandchildren” and “give attention to older parents”, they may feel more integrated and useful, thus preventing their cognitive decline.

## 5. Conclusions

In China, older parents have become an important source of childcare for their grown children since 2010. However, caring for babies may affect older parents’ psychological health (PH) in both positive and negative ways. Using stepwise decreasing logistic regression, this study aimed to assess the factors associated with the PH among caregiving older parents (400 respondents) based on the public panel data of the China Health and Retirement Longitudinal Study (CHARLS). The findings showed that being male (X1, OR = 1.661 (95% CI 1.066–2.590)), being literate (X4, OR = 2.129 (95% CI 1.369–3.309)), and expecting long-term care in the future from their grown children (X6, OR = 2.695 (95% CI 1.736–4.185)) were significant factors associated with PH. Therefore, in such an aging society, we should not simply regard our older parents as a “burden”; we should recognize and appreciate their contribution to caregiving. In the meantime, it is important to give older parents the necessary economic and emotional support to maintain their psychological health.

## Figures and Tables

**Table 1 ijerph-17-00556-t001:** Description of variables.

Variables	Criteria of Dependent Variables	Index Description and Scoring in Regression	*N*: Total Respondents	Value = 0 (*n*/%)	Value = 1 (*n*/%)
Dependent Variable	Good psychological health	Excellent, very good and good = 0, poor and very poor = 1	400	237 (59.2%)	163 (40.8%)
Independent Variables (X)	Gender (X1)	Male = 0, Female = 1	400	179 (44.7%)	221 (55.3%)
	Age (X2)	Age under 65 years old = 0, Age over 65 years old = 1	400	153 (38.3%)	247 (61.7%)
	Residence (X3)	Live in the urban area = 0, Live in the village = 1	400	89 (22.2%)	311 (77.8%)
	Educational status (X4)	Literate (Primary school or above) = 0, Illiterate = 1	400	194 (48.5%)	206 (51.5%)
	Marital status (X5)	Married = 0, Another situation = 1	400	308 (77%)	92 (23%)
	Expectations of long-term care from their grown children in the future (X6)	Yes = 0, No = 1	400	263 (65.7%)	137 (34.3%)
	Educational status of grown children (X7)	Literate (Primary school or above) = 0, Illiterate = 1	400	58 (14.5%)	342 (85.5%)
	Living place of grown children (X8)	Live with their older parents = 0, Not live with their older parents = 1	400	167 (41.8%)	233 (58.2%)
	Marital status of grown children (X9)	Married = 0, Another situation = 1	400	318 (79.5%)	82 (20.5%)
	Physical health of grown children (X10)	Excellent, very good, and good = 0, poor and very poor = 1	400	241 (60.3%)	159 (39.7%)
	Economic status of grown children (X11)	At least own one house = 0, No house yet = 1	400	208 (52%)	192 (48%)

**Table 2 ijerph-17-00556-t002:** Logistic regression results in the first step.

Good Psychological Health	OR	S.E.	Z	*p* > |Z|	95% CI	
					Lower	Upper
Gender (X1)	1.828 **	0.443	2.49	0.013	1.137	2.940
Age (X2)	1.083	0.286	0.30	0.761	0.646	1.817
Residence (X3)	1.532	0.430	1.52	0.129	0.884	2.656
Educational status (X4)	1.810 **	0.442	2.43	0.015	1.122	2.921
Marital status (X5)	1.029	0.279	0.11	0.915	0.605	1.753
Expectations of long-term care from their grown children in the future (X6)	2.707 ***	0.614	4.39	0.000	1.735	4.222
Educational status of children (X7)	0.939	0.230	−0.20	0.845	0.503	1.755
Living place of children (X8)	0.927	0.220	−0.32	0.749	0.582	1.476
Marital status of children (X9)	0.653	0.210	−1.33	0.184	0.348	1.224
Physical health of children (X10)	1.496	0.335	1.80	0.072	0.965	2.320
Economic status of children (X11)	1.165	0.286	0.62	0.536	0.719	1.886
Constant	0.159	0.082	−3.55	0.000	0.058	0.438

OR, odds ratio; S.E., Standard error of the coefficient; Z, Z statistics; CI, Confidence Interval; ** *p* ≤ 0.05; *** *p* ≤ 0.001.

**Table 3 ijerph-17-00556-t003:** Logistic regression results in the second step.

Good Psychological Health	OR	S.E.	Z	*p* > |Z|	95% CI	
					Lower	Upper
Gender (X1)	1.836 **	0.439	2.54	0.011	1.149	2.933
Age (X2)	1.090	0.281	0.33	0.739	0.657	1.808
Residence (X3)	1.531	0.429	1.52	0.129	0.883	2.654
Educational status (X4)	1.816 **	0.440	2.46	0.014	1.129	2.921
Expectations of long-term care from their grown children in the future (X6)	2.705 ***	0.613	4.39	0.000	1.734	4.219
Educational status of children (X7)	0.939	0.299	−0.20	0.845	0.503	1.755
Living place of children (X8)	0.925	0.219	−0.33	0.741	0.582	1.470
Marital status of children (X9)	0.653	0.209	0.133	0.184	0.348	1.224
Physical health of children (X10)	1.499	0.335	1.87	0.07	0.967	2.321
Economic status of children (X11)	1.162	0.285	0.61	0.540	0.719	1.879
Constant	0.159	0.082	−3.55	0.000	0.057	0.438

OR, odds ratio; S.E., Standard error of the coefficient; Z, Z statistics; CI, Confidence Interval; ** *p* ≤ 0.05; *** *p* ≤ 0.001.

**Table 4 ijerph-17-00556-t004:** Logistic regression results in the third step.

Good Psychological Health	OR	S.E.	Z	*p* > |Z|	95% CI	
					Lower	Upper
Gender (X1)	1.824 **	0.432	2.54	0.011	1.147	2.900
Age (X2)	1.094	0.282	0.35	0.729	0.660	1.811
Residence (X3)	1.538	0.430	1.54	0.124	0.889	2.662
Educational status (X4)	1.835 **	0.434	2.56	0.010	1.154	2.918
Expectations of long-term care from their grown children in the future (X6)	2.705 ***	0.613	4.39	0.000	1.734	4.220
Living place of children (X8)	0.921	0.217	−0.35	0.728	0.580	1.463
Marital status of children (X9)	0.650	0.208	−1.35	0.178	0.347	1.217
Physical health of children (X10)	1.498	0.335	1.81	0.07	0.967	2.321
Economic status of children (X11)	1.168	0.285	0.64	0.524	0.724	1.883
Constant	0.150	0.062	−4.56	0.000	0.066	0.339

OR, odds ratio; S.E., Standard error of the coefficient; Z, Z statistics; CI, Confidence Interval; ** *p* ≤ 0.05; *** *p* ≤ 0.001.

**Table 5 ijerph-17-00556-t005:** Logistic regression results in the last step.

Good Psychological Health	OR	S.E.	Z	*p* > |Z|	95% CI	
					Lower	Upper
Gender (X1)	1.661 **	0.376	2.24	0.025	1.066	2.590
Educational status (X4)	2.129 ***	0.479	3.36	0.001	1.369	3.309
Expectations of long-term care from their grown children in the future (X6)	2.695 ***	0.605	4.42	0.000	1.736	4.185
Constant	0.241	0.052	−6.65	0.000	0.159	0.367

OR, odds ratio; S.E., Standard error of the coefficient; Z, Z statistics; CI, Confidence Interval; ** *p* ≤ 0.05; *** *p* ≤ 0.001.

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
