# Peer review of "Factors Associated with the Psychological Health of Caregiving Older Parents and Support from Their Grown Children: Results from the China Health and Retirement Longitudinal Study"

_ijerph, 2020, doi:10.3390/ijerph17020556_

Round 1
Reviewer 1 Report
This is an interesting study, but I suggest an extensive change of the article, namely in background, discussion and conclusion. The authors's perspective are not convergent with the new evidences in the field. Usually, evidences show that there are health benefits of grandchild care, and that this activity are important given the widespread provision of grandparental childcare in countries. I can understand that authors can find different results, but in this case a strong explanation is needed, underlying mechanisms and causal pathways between grandchild care and grandparent health (burden).
Authors can find my suggestions on the PDF. I'm free to help, this is an important topic.

Author Response
Dear reviewer,
Thank you, thank you so much for your approval, your constructive comments and your kind help!
Merry Christmas!The response, Please see the attachment.
corresponding author: wangyuhui

Reviewer 2 Report
The paper attempts to summarize the contribution that specific factors make on self-defined psychological health. Although limited by the variables available in the secondary data, the paper develops a model that shows the contribution certain variables make to grandparents' psychological health. Although the paper makes some contribution to identifying negative consequences of grandparenting, there was in fact NOTHING about grandparenting. Missing from asserting that this is related to grandparenting are the relationship with grandchildren, caregiving frequency and duration and caregiving pressures, number of grandchildren, the age dispersion of grandchildren, living proximity of grand children, gender of grandchildren. These variables were missing from the analyses that might suggest a limitation of the database used. Also the paper, when it showed some contribution to the psychological health, such as with "grandparents expect to get long-term care in the future” and “respondents’ grown children in good health” these were not explored further, either through literature review or through further analysis of the database or other databases.
Specific grammar and typos
30 "imperfections" replace with "inadequacies"
37 delete the rest of the sentence “development…educational theories” this is an orphan statement and does not make sense
38 edit to read “not only do they need to take care”
46 include “a” in front of favour and delete “the way of”
49 replace “analysis” with “literature”
50 after grandparents include,”, either positive or negative effect.”
56 insert “initially” before operated and delete “at first”
59 replace “then” with “the”, “ethical proof” with “Institutional Review Board review”
61 replace “included” with “selected”
66 variables (plural) and then replace “(which” with “(those” replace ‘fallacy” with “error”
67 delete F2F (factor to factor) and replace with “Multicollinearity where very high intercorrelations or inter-associations among the independent variables”
69 replace “factor” with “variable” and delete “certified as”
70 delete “. Ant it would be certified” and replace with the continuation of the same sentence “and”
71 delete “between 0 and” with “below”
Table 2 and 3Expectations of Long Term Care has three asterisks without a corresponding explanation below (*** <p=0.001)
We don’t report p=.1 this alpha value is rarely used.
113 delete “too”
Tables 5, 6, 7, 8 and 9 the heading is “physical health” do you mean “Psychological Health”?
All these different iteration of the model do very little to explain the data. I would delete tables 5, 6, 7, and 8 and leave 9 on its own.
152 delete “social” and replace “provided” with “are providing”
155 replace whole sentence with “Results showed that being male and educated were protective factors.”
156 it could also be that perhaps men do not do as much caregiving as women and hence why they report better PH. Education has other confounds that include better employment, higher income, better pensions and better living conditions.
References
[1] [20] missing date
[27] on suicide delete as it does not apply
Author Response

(The authors gave the same response as above.)

Round 2
Reviewer 1 Report
The article is now much better and has incorporated the proposed amendments. I leave in the PDF some minor suggestions to improve the text and finalize the document

Author Response
Dear reviewer,
Thank you! Thank you for your approval and support!
Thank you very very much!
For the response, please see the attachment.
Happy New Year's Day!
Yours sincerely,
wangyuhui

Reviewer 2 Report
I appreciate the author/s attempt to address the original criticisms. Two of the major shortfalls remain unaddressed.
The concept of grandparenting assumes that there are grandchildren. Although the authors acknowledge the original criticisms---by defining these as a limitation in the study---such admission does not eliminate the fact that you cannot write about grandparenting without addressing the relationship with their grandchildren. If the title of the study was Older Parents, then this would be correct as you include data on their children. But since there is nothing in the analyses that include grandchildren this study remains limited. This shortfall denies this work to be published by itself. The ten logistical regression tables need to be summarizes into a coherent table rather than reported individually. There is no advantage to reporting each results independently when the conclusions are basic.Author Response
Dear reviewer,
Yes! We have known the major mistakes about what we studied in this paper.
For the important two points, we have successfully modified them. Point-by-point response, please see the attachment.
With your help, we made it right!
And you also gave us the right dirction and idea of research in the future!
Thank you, thank you so much!
Heppy New Year's Day!
Yours sincerely
wang yuhui
